# Developing an Embedded Nursing Service within a Homeless Shelter: Client’s Perspectives

**DOI:** 10.3390/ijerph18094719

**Published:** 2021-04-28

**Authors:** Denise Warren, John Patrick Gilmore, Christine Wright

**Affiliations:** 1School of Nursing, Midwifery and Social Work, Canterbury Christ Church University, Canterbury CT1 1QU, UK; christine.wright@canterbury.ac.uk; 2School of Nursing, Midwifery and Health Systems, University College Dublin, D04 V1W8 Dublin, Ireland; john.gilmore@ucd.ie

**Keywords:** homeless, nursing, nurse-led, shelter, healthcare

## Abstract

This phenomenological case study of a newly developed nursing service, embedded within a homeless shelter in the South East of England, uses semi-structured to elicit experiences and perceptions of clients within the service. Participants (*n* = 6) were interviewed using a semi-structured approach and identified three broad themes: impact of previous healthcare experiences, benefits of embedding healthcare within the shelter, and future service development. The study illuminates the diversity and complexity of healthcare needs of homeless people, as well as offers a unique insight into the service user’s perception of the service.

## 1. Introduction

Homelessness is an increasing social issue globally [1], and rates of homelessness in England remain high; while fluctuations are reicorded between periods, overall, there has been a marked increase in homeless figures across the last decade [2]. An increase in the overall homeless population along with an ageing homeless population [3] bring the health inequalities faced by this group into focus. There are numerous specific health inequalities experienced by the homeless population; with disproportionate experiences of both physical and mental illnesses, harmful substance use, and long-term chronic illnesses compared to the housed persons [4,5,6,7,8]. In a review of homeless persons’ experiences of health and social care, Omerov et al. [9] illuminated the diversity of systems of provisions and experiences globally. There is no agreed best practice in terms of service delivery type, however it is agreed that targeted provision of health and social care to homeless communities is warranted. The NHS England Long Term Plan [10] focusses on primary care integrated care systems as the main driver to tackle health inequalities; however Batchelor and Kingsland [11] note the necessity to deal, not only with the service delivery framework, but also on the wider causes of ill-health amongst homeless populations. Roche et al. [12] identify the utility of nurse-led services as a key enabler around preventative healthcare for homeless populations; and Poulton et al. [13] identify public health nursing as a key role fit, not only in intervention but also in the assessment of need, self-care skills to meet need, referral and access to further care, partnership working, health promotion and protection as well as engagement in policy influence, and strategy development. The Queens Nursing institute homeless health project started in 2007 to support UK nurse’s development and learning needs to gain knowledge in how to deliver care to the some of the most marginalised section of the population including the homeless. Their aim to improve healthcare for the most vulnerable and hard to reach patients; however, no clear model of delivery is proposed. As highlighted, in the NHS Long Term Plan [10], primary care is seen as a key driver of health and healthcare delivery to dismantle health inequalities, however upwards of one third of homeless people in the UK are not registered with a primary care provider such as a GP [14].

The service explored in this project stemmed from an initial one nurse weekly voluntary service within a homeless shelter in the South East of England, to a two nurse in-shelter clinic. The shelter runs a day service drop-in with provision of meals, a clothes bank and washing facilities and also has a winter emergency weather response programme which provides on-site accommodation throughout the winter months. There were 12,635 separate visits to the shelter during this year, with 386 individuals accessing services. The nurse service included general health screenings with weight, height, blood pressure and pulse measurement and recordings as well as recording of past medical history and medicines information. Simple treatment and dressings for uncomplicated wounds were also provided and for more complex health issues, service-users were assisted with making appointments, referred to emergency GPs and accompaniment to clinics and the emergency department. A total of 50 individuals accessed the nursing clinic on at least one occasion over the year. Given that homeless voices are often silent in conversations around homeless health services this project centres the homeless persons experience.

## 2. Materials and Methods

A phenomenological case study approach was adopted. The lived experiences of those accessing the service were paramount and through participant semi-structured interviews an in-depth understanding of the meaning of this embedded nursing service within the homeless shelter was illuminated. The relevance of using a case study method is a more practical approach as the questions posed will require extensive and ‘in-depth’ description of the phenomenon [15], the phenomenological underpinnings centres and gives primacy to the lived experiences of research participants [16].

A purposive sampling strategy was adopted, whereby information sheets were available throughout common areas in the shelter and clinic attendees informed of the study when they attended. Six frequent attendees of the shelter were recruited following informed consent processes where the information sheet was read with the researcher and any questions attended to. All of the participants had been street sleeping for a number of years, (2 years to 11 years) and had been born in either Ireland or the UK and were currently living in the local area. Two women and four men were recruited. The inclusion criteria were that participants be:Homeless described as street living, living in temporary accommodation or with no permanent housing (hostel, sofa surfing)In full agreement and understanding that all interviews would be recorded, transcribed, member checked with full anonymity.Fully fluent in English speaking and understanding.Not demonstrating signs of being under the influence of alcohol or drugs at the time of interview or during the member checking.

The purposive sampling strategy allowed for researchers to ensure a broad representation of age and gender attendees of the service was represented.

The research took place over six months in 2017. A number of semi-structured interviews took place with participants within the shelter, this was accompanied by reflexive diary notes for data analysis purposes.

Researcher positionality was key within this study, with the lead researcher occupying an insider-outsider position. While the conducting of research positioned the lead researcher in a different to usual role and therefore an outsider; the insider role of volunteer nurse within the service had several benefits, including the identification of appropriate research questions. Familiarity with the service meant that the researcher more likely to understand the specificities of participants’ descriptions of their experiences [17]. Access was also made much more accessible by having an already built up relationship with the research participants; leading to a greater openness to engaging with the research and willingness to share their experiences [17,18,19]. Obviously caution and consideration was given to ensure that previous relationships did not impact on the research findings in terms of bias; this was done through ensuring that reflexivity was incorporated throughout the data collection, analysis and writing up of the research [18,20,21].

The reflexive approach taken incorporating diary inputs with the interview data ensured that the researchers could consider their own preconceptions prior to, and during data collection and analysis [22]. The reflexivity and acknowledgement of subjectivity in the researchers’ positionality, enhanced the research process by bringing awareness of things that could contribute to the study findings [23].

All interviews were audio recorded and verbatim transcriptions created; the full transcriptions were then reviewed by participants in advance of analysis to ensure an accurate representation of experiences [24]. All the participants stated that they were satisfied with the transcribed interviews and the views and statements were unaltered. Participant information was then anonymised with codes related to gender and age.

Transcribed interviews were uploaded to NVivo for coding and initial organisation; following initial organisation and reflexive diary entries were subsequently considered [25] when organising statements into significant themes and subthemes using Colaizzi’s data analysis process [16] with final validation of findings from the research participants (see Figure 1).

## 3. Results

Participant experiences illuminated three broad themes for consideration: impact of previous healthcare experiences; benefits of embedding healthcare within the shelter; and future service development.

### 3.1. Impact of Previous Healthcare Experiences

Throughout the interviews it became evident that many of the clients experienced difficulties when asking for help for their healthcare needs; they expressed feelings of contested worthiness, rejection and loss of confidence. One man explained:

My first time for asking for help medically was horrible, when I had my mental health breakdown two years ago. I had never had a problem with my mental health before, never ever. I’d got a bit depressed before but this time I ended up getting sectioned because I was on the streets and I kept asking for help and getting, no, no, no, no, no, no, no.(M, 49)

Healthcare interactions often left participants feeling dehumanised and ‘worthless’. One man described how when he did visit the GP he felt that the doctor avoided eye contact and didn’t listen to him:

Like I’ve presented before with symptoms and because I’m homeless his attitude was, just not listening to me, making notes on his computer, not looking at me, not speaking to me properly to try to find out what was wrong with me, what was causing the problem.(M, 54)

He went on describe how this experience had a major impact on future health seeking behaviour and although he has a life-threatening condition, he was reluctant to seek medical assistance until he visited the nurse-led clinic at the shelter, the nurse offered an interface between him and mainstream healthcare:

When you phoned up when I first came to see you, you phoned and booked me a doctors’ appointment, you remember? Now the fact it was you dealing with that receptionist rather than me, I would have given up and would have been fobbed off. You made sure I saw another doctor, you said that’d alright as you knew I couldn’t see that doctor again. I wouldn’t have gone back again but I knew I could see another doctor and it incentivized me.(M, 54)

Participants discussed feeling unwanted within healthcare environments. One participant described a presentation to the emergency department with pneumonia and fractured ribs:

Later about fiveish we went to a little ward bit and they said I could stay the night cos I had pneumonia and cracked ribs cos of the coughing. About sevenish a doctor came and said I could go home. I told him I was told I could stay the night but he said I was discharged. I told him I was staying in a tent and he said not to get wet. I had to walk back to the tent from the hospital. They gave me some pills and that was that. I’m used to it but I haven’t felt that rough before.(M, 67)

Another participant described the experience of rushed treatment, a hesitancy for healthcare providers to engage in any meaningful ways with her:

I took some heroin and passed out so they called an ambulance. I don’t take heroin but I was feeling shit and she gave it to me. I hit my head cos it was in a toilet down the town. They gave me an injection when I got there to bring me round and before I know it, I’m out the door. I was feeling bad and was unsteady and that but they told me I couldn’t stay there.(F, 54)

Participants gave several accounts of their experience of the stigma and stereotyping surrounding homelessness and how it impacted on their healthcare encounters:

The most frustrating thing for me is because I’m reasonably intelligent I know that some nurses and some doctors jump to conclusions, that I fit into this pigeon hole or that pigeon hole which then has an effect on how I’m treated by the rest of the staff, which is horrible.(M, 49)

Another participant describes how a disclosure of homeless abruptly changes the relationship between healthcare professional and patient:

As soon as they know you’re on the streets you can see their face change like I stink of something. They just talk to you badly and don’t bother with you no more.(F, 54)

There was a perception by participants that generalisations were made about them because they were homeless, this one aspect of their identity then impacted on their access to appropriate care:

They don’t care about us lot. I know what they think that I drink and take drugs and that. I don’t though, you know this. I only smoke tobacco and have a lager sometimes. I know I look bad and that but I lost everything. I’m in my sixties and I don’t have anything.(M, 67)

He had premeditated ideas of who and what I was. He asked how long I’d been on heroin? I told him I have never used heroin as I know I would love it, so I chose not to try it. He then asked when I had had my last drink today and I said I don’t drink; I do not drink at all. I said you’re mistaking me for someone else or you’re making these assumptions about me.(M, 49)

People will judge me everywhere, doctors, nurses, chemists and that’s the truth.(W, 32)

Negative experiences were linked to the development of internalised negative perceptions of homelessness and other homeless people by participants:

Some people here cause loads of problems, I mean, you’ve seen the people here, some people are ok to talk to and others don’t want to talk. Some of them want to fight, they blame everyone else. Instead of blaming themselves they blame other people. But the point about the health service is it must treat everyone whether you’re rich or poor or whatever and that was the creation of the health service.(M, 59)

My problem being homeless is mine, it’s my fault and that’s how it is. Pride is not a good thing, I know but it’s my problem and I can’t give it to anyone else to hold(M, 54)

There were narratives of division between types of homeless people, and a distancing from participants from a certain ‘type’ of homeless person:

A lot of the people here get a hard time I know but they don’t look after themselves. They aren’t like me and keep themselves clean. I don’t drink and a lot of them do. I think that people think badly about the homeless and this can cause problems with the help they receive.(M, 54)

When I leave here I go to the library. I do crosswords and write. That’s my day. Some of these people can’t read or write and turn to alcohol. If I couldn’t read or write, I would probably drink or take drugs.(M49)

Participants noted that healthcare professionals have little interest in the wider social, emotional and mental health of homeless patients or of the complex intersections between mental ill health and homelessness:

No ones bothered about mental health. They haven’t got mental health problems like a lot of the homeless so they’re not bothered about mental health people. Unless something happens to them or their family they will find out there’s no place for them to go. There used to be places in the 60′s, 70′s and 80′s, massive places that you could go to for help. That’s why there’s so many people on the streets.(M, 59)

The depression is bad and manifests itself into physical pain, phantom pains which I know are associated with my depression. I get really low sometimes and to have people just dismiss it, to show no understanding of that is just terrible, it’s the worst thing ever.(M, 49)

My Father raped me as a kid and I lost two of my babies. No one helped me really so I sorted myself out. Not in the right way I know. I drink and stuff so I don’t think about all that shit. My heads fucked but I’ve just got to get on with it.(F, 54)

There were however some positive experiences of healthcare encounters that participants shared, access being highlighted as a particular important aspect.

Most of the time the doctors have been good. Getting appointments, everything really, well most of them [laughs].(M, 54)

I’ve had no problems with the health service as in seeing someone. I didn’t see a doctor for 7 years but when I did decide to see one I got straight in and he sorted my medical problems that I’ve got.(M, 59)

### 3.2. Benefits of Embedding Healthcare within the Shelter

The non-judgemental approach of embedded nursing staff was seen as significant benefit to the services. While limitations of the service were acknowledged, the nurses were seen as key links between the homeless service users and mainstream healthcare:

You can’t get everything here but just having someone to listen to you is good and you can advise them, give them advice about medical stuff and being homeless, the main things are being cold, chest problems you can help them to get help.(M, 59)

As soon as you are here you let people know you’re here. It’s very relaxed. Both of you are nice and that opens up healthcare. The homeless don’t trust many people because they aren’t able to work people out.(M, 49)

Sometimes you are trying to sort out the doctors but it’s too early especially at the moment. I have to move my stuff out of the church and it takes time. I can’t get to the doctors on time so it’s good that you’re here.(F, 45)

I think its good. You’ve sorted some bits out for me and there ain’t many places that have this.(F, 32)

Having someone do your feet is like a treat. My feet felt better after seeing her and she’s nice too.(M, 54)

I think the new foot lady is an excellent idea. A lot of us have problems with their feet cos of the walking and cold.(M, 59)

The open and non-judgemental approach of the nurses in the service was noted as an enabler to access but also in building therapeutic relationships:

You two girls are proper gems and nice with it. I can talk to you about anything and you check my blood pressure if I ask and it’s good you come here. We all know you don’t get paid and so you must like us lot [laughs].(F, 54)

You two don’t look down on me and I know what I’m like. You try and help us lot here and I know I can talk to you about anything really. You always come and say hello and stuff.(F, 54)

I know I can come and tell you anything. You are both the same, I know I can tell you anything like I had sex with a hedgehog last night and I know there’s no judgment. [laughs] That’s such a big thing.(M, 49)

I’ve got terrible feet and she didn’t bat an eye lid. I was really embarrassed but she didn’t say anything and my trotters feels much better.(F, 54)

You don’t feel out of place coming in here and you never say you can’t see me. I know I missed the doctors the other day and I know they’re probably pissed off. I’d rather come in here.(F, 32)

Convenience was seen as a clear benefit to the nurse-led clinic, having at least some service on-site and staff willing to care:

Having a nurse say its important or this needs to be done quickly makes a massive difference, it makes you sit up and think, it give you the right and sometimes I don’t think I have the right.(M, 49)

I think that people can come in here, shut the door and you may not cure what they’ve got but they can chat to you.(M, 59)

You both have the time to listen and you both have the right attributes to do this as well.(M, 54)

Knowing there’s no queue, I don’t have to book in advance and even the simple thing of holding my meds for me, that makes a difference.(M, 49)

I don’t have to make an appointment, I can have my breakfast and then pop in [laughs].(M, 67)

### 3.3. Future Service Development

Overall, participants were hesitant to provide any critique of the service, or suggestions for service improvement. Participants were very keen to express gratitude for the service as it was. There was a suggestion that medical intervention could provide for a more holistic and extensive service

I can’t think what else you can do unless you had a doctor here who could prescribe stuff. If we had a doctor he can do a lot more things than you like prescribing drugs and sending people to specialist Doctors if they need it.(M, 59)

I think having a doctor would be good, yeah a doctor to give prescriptions.(F, 32)

If you could give out drugs and that, like antibiotics and stuff for pain. I know you said this maybe happening but it would be good now so we don’t need to bother the hospital.(F, 54)

The service was seen as a useful draw to engage homeless people with healthcare and it was felt that provision of broader services may engage more people:

It would be better if you was here more than you are(M, 67)

I am very happy with the service you give but some of the others here need special people to come in such as more mental health people, drug issues, drinking. I think specialists should come in. But I’m very happy with the service here and what you give us. Its convenient.(M, 54)

I think it would help if the centre had a doctor available. It would be a continuation of what you do and I think a lot more people would come in too. Some come in on fleeting visits, if there was a doctor here it would make a difference because there are people here that need to see a doctor. When you are feeling low you can’t cope with dealing with a doctors’ surgery, having one here would be so beneficial.(M, 49)

### 3.4. Description and Structure of the Phenomenon

Experiences of accessing and engaging with an embedded nursing service in a homeless shelter were illuminated through interviews with the six participants in this study. An overarching positive experience was described by all participants and the role of the service in building on their health consciousness and perception of healthcare and healthcare professionals was explored. Participants’ shared narratives of their previous encounters in healthcare environments which had a negative impact on their perception of healthcare and healthcare professionals; as well as on their own health and health-seeking behaviours. The embedded service was seen to facilitate a building of trust and familiarity with the nurses working there. Participants discussed their comfort in accessing the service and working there, the lack of negative judgement experienced and the role of the service in bridging their access and health-seeking behaviours in other areas of healthcare.

## 4. Discussion

The experiences of negative healthcare encounters illuminated through this phenomenological case study concur with other studies examining homeless people’s experiences of healthcare and impact on health [26,27,28,29].

It is evident that homeless people will, as result feeling stigmatised and rejected, attempt to maintain their positive sense of self by protecting themselves. They do this by separating themselves from the perceived rejection, discrimination and disapproval [30]. Those interviewed in this study clearly articulated how experiences of rejection and discrimination had a profound impact on their health-seeking behaviours.

Many of them felt this was due to assumptions, to preconceived ideas that are formulated about homeless people, for example that they drink and take drugs. From the six interviewees, four expressed that they did not in fact use drugs and may occasionally socially drink and having assumptions made regarding this made them feel further stigmatised.

Not only are access and behaviour issues impacted by this sense of stigma and discrimination but too, when these experiences are encountered in healthcare settings, feelings of mistrust of healthcare professionals are built up, further impacting on access [31].

The participants in this study discussed feelings of being unwelcome within healthcare environments, that they felt rushed and their own healthcare perspectives dismissed by healthcare professionals; this feeling of disregard further alienates the homeless person and is a significant barrier to care. Feeling listened to can have a profound positive impact on healthcare encounters for homeless people [32,33]. Simple processes such as therapeutic listening can avoid these feelings being build up, just listening allows healthcare practitioners to not only establish the needs of the homeless person they are caring for but too, to advocate more effectively for their healthcare needs.

The advocacy and signposting role of the nurse was highlighted as significantly important for the participants in this study; there was a concern for many that their lack of knowledge around institutional processes would further stigmatise them—some of these institutional barriers and perceived institutional barriers are some of the most significant issues around health-seeking behaviours [34]. People affected by social exclusion may lack the voice, the skills or the stamina to raise issues and advocate for themselves. This was evident as many of the participants expressed a need for support, even for the simplest tasks, such as making a GP appointment. Listening to homeless people within healthcare not only ensures more appropriate treatment, but also presents an opportunity for structural change [26].

The embedding of the nurse-led clinic within a homeless shelter was a key enabler around healthcare access for the participants in this study—their positive evaluation of the service not only referred to the convenience of having healthcare provided within a service they already access, but also the way in which the service was provided. Accessing services can be challenging for the homeless population and barriers to healthcare for this populace are attitudinal and structural; with inflexibility within the organisations’ very often leading to discrimination [35]. Having nurses work within the service provided a more culturally conscious level of care—it is vital that when a homeless person seeks healthcare, that the advice is personally pertinent and takes into account the person is not living a ‘housed life’ [36]. Some of the participants in this study expressed that they had felt dejected through previous healthcare experiences where advice had been inapt and unrealistic for their life situation. Awareness of the specific issues facing homeless people allows for a more in-depth understanding of the healthcare issues and inequalities and therefore facilitates more client-responsive interventions [37]. The embedded nature of the nurse-led clinic was seen by participants to take away some of the pressure associated with health-seeking, the did not feel obligated to access any particular services provided but knew they were available if needed. If services are offered voluntarily, not obligatory, they are likely to support social attachment and decrease isolation amongst the homeless populace [38].

There was a desire for further development of the services provided articulated by participants within the study, however the feasibility of an embedded multidisciplinary healthcare service within a single shelter could be contested. Primary care is absolutely essential in addressing the multifaceted healthcare needs of the homeless population [39] and with the drive for Integrated Care Systems now being implemented in the UK [10], embedded nurse-led services could play a key linking role. Nurses not only have a large scope in terms of interventions, but the role of nurses is assessment, planning and referral make them a sensible fit to lead these services [14].

## 5. Conclusions

Homelessness in itself brings a multiple of layers of discrimination, stigma and social exclusion; illuminated in this study was how those experiences and perceptions impact on how a small group of people experiencing homelessness view and access health services and how they perceive the nurse-led clinic within the homeless shelter they use.

While not a formal evaluation of this service, the narratives of those who use it give testament to the value of embedding nurse-led services within homeless support.

### 5.1. Limitations

Given the small sample size and single site setting of this research project generalisibility of the findings is not possible; transferability or comparability around similar groups or similar contexts is not the aim, however. The findings of this project need to be understood in relation to homeless participants who took part, and the analysis of their experiences understood in relation to the positionality of the primary researcher as their nurse. The experiences are however useful in provoking thought and reflection for practitioners and policy makers involved in the area of homeless health or inclusion health.

### 5.2. Final Thought

Participant voice is the central aspect of this study—Finally, in conclusion a poem written by one of the participants is presented. He chose the pseudonym ‘Sean Street’ and gave written permission for to use this within my study.
BelieveNow as I sit on the street alone again, trying to avoid the cold and the dreaded rain.I start to wonder where it all went wrong, I used to be smart, I used to be strong.Having it all has never been my goal, but neither was being homeless and on the dole.Life can go wrong in the blink of an eye, I always knew that, but I’m still asking why?Is it something I can put down to fate, or can I change it, before it’s too late.I know to get through it, I have to be strong, so I have to stop focusing on where it went wrong.Of course I should learn from my mistakes, but I need to focus on getting better, and doing all that it takes.First is make sure I start looking after myself, not just my physical, but my mental health.Start making notes of what is important to me, to guide me to a place where I want to be.I don’t want much, just to be in a better place, and feel like I’m a part of the human race.To be able to function and give something back, is a definitive sign that I’m back on track.The hardest part will be to get off the streets, so remember the wins, forget the defeats.There’ll be many obstacles to get me depressed, I need to stay focused, and not get stressed.This all sounds simple, perhaps I’m being naive, but the thing that matters most, isI’ve got to believe.(Sean Street, 2016).


## Figures and Tables

**Figure 1 ijerph-18-04719-f001:**
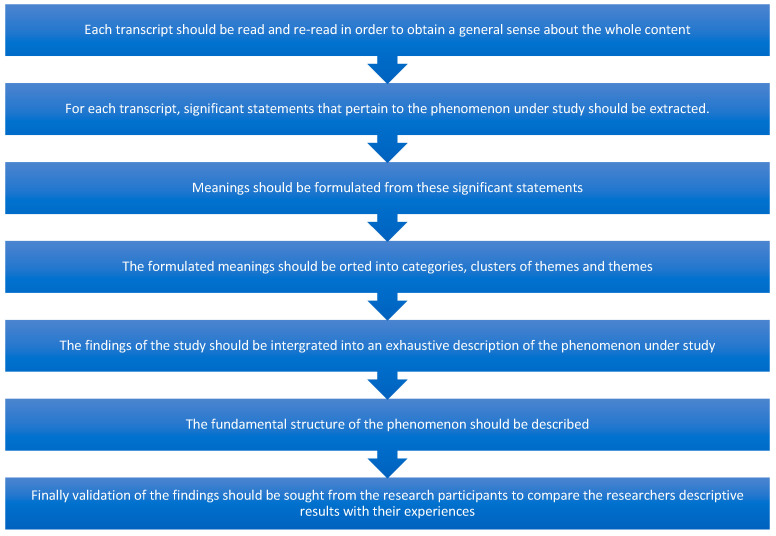
Collazi’s Process for data analysis.

## Data Availability

Data sharing is not applicable to this article.

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
