# Peer review of "Developing an Embedded Nursing Service within a Homeless Shelter: Client’s Perspectives"

_ijerph, 2021, doi:10.3390/ijerph18094719_

Round 1

Reviewer 1 Report

This manuscript reports on a very small sample of homeless individuals who used a nursing service embedded within a shelter program.  The contribution of the manuscript to the field is not strong because of the limited sample, potential bias given that the clinical provider is also the researcher, and lack of contextualization of the service within a field where there is a fair amount of knowledge and model development- at least in the US where there is a large and extensive Health Care for the Homeless program delivering health care services on the street, in shelters, and in free standing clinics.  The manuscript could be improved if the authors provided a better description of the actual services delivered and the size of the population served from which the N=6 sample was drawn, a demographic description of the small sample chosen, as well as their length of relationship and frequency of service use from the nurse. Some independent coding of the transcripts would give more weight to the themes which appear to be developed primarily by the nurse provider.  In addition, the introduction should cite manuscripts that describe other embedded homeless health care programs briefly the reasons for creation of such programs, and the state of the field in the UK.  It appears, but is not clear, that providing health care at shelters is not common in the UK and that this was an informal, volunteer service but this needs clarification.

Author Response

Dear Reviewer,

Many thanks for your consideraion of our research project based on the experiences of clients' of an embedded nurseing service within a homeless in shelter in the UK.

In responding to your generous feedback:

1) We have provided more context around the provision of homeless healthcare in the UK, establishing that there is not preferred or suggested best-practice model.

2) We have given further information about the service itself and demographics of the participants.

3) We have produced a clear limitations section 

As suggested by the other reviewer some further revisions were made

1) We have reframed the methods as a phenomenological case study which is more inkeeping with the methods applied. As suggested by Yin (2018) this design allows for a contemporary phenomenon to be understood within its real-life context; when the boundaries between phenomenon and context are not clearly evident. While not strictly phenomenology, the centring of participant experiences and use of a phenomenological process of data analysis; this was the most appropriate framing, a case study with phenomenological underpinnings. There was a tidying up of some of the language around 'field notes' which were in fact reflexive notes used to enhance reflexivity around the data analysis process.

2) A clear description and structure of the phenomenon illuminated is provided in the findings chapter in-keeping with the data analysis process

3) Grammatical and syntax edits were completed as suggested the article will be proof edited prior to publicaiton. 

Many thanks once again for taking the time to review this article. 

Reviewer 2 Report

This is an interesting piece of work forwarding the voices of homeless people using an embedded nursing service. The main difficulty I see is the framing of the study. As noted in my comments it is variously referred to as a case study, anthropology, phenomenology, etc. Naturalistic inquiry appears apt. It is definitely NOT phenomenology.  If it is anthropology, where are the observations used? Where is the description of the settings, scene, and researcher input? The author(s) need to settle this and align subsequent sections within the chosen frame. 

An edit for sentence length, missed punctuation, etc. is also needed.

Author Response

Dear Reviewer,

Many thanks for your consideraion of our research project based on the experiences of clients' of an embedded nurseing service within a homeless in shelter in the UK.

In responding to your generous feedback:

1) We have reframed the methods as a phenomenological case study which is more inkeeping with the methods applied. As suggested by Yin (2018) this design allows for a contemporary phenomenon to be understood within its real-life context; when the boundaries between phenomenon and context are not clearly evident. While not strictly phenomenology, the centring of participant experiences and use of a phenomenological process of data analysis; this was the most appropriate framing, a case study with phenomenological underpinnings. There was a tidying up of some of the language around 'field notes' which were in fact reflexive notes used to enhance reflexivity around the data analysis process.

2) A clear description and structure of the phenomenon illuminated is provided in the findings chapter in-keeping with the data analysis process

3) Grammatical and syntax edits were completed as suggested (many thanks especially for taking the time to provide these) the article will be proof edited prior to publicaiton. 

As suggested by the other reviewer some further revisions were made

1) We have provided more context around the provision of homeless healthcare in the UK, establishing that there is not preferred or suggested best-practice model.

2) We have given further information about the service itself and demographics of the participants.

3) We have produced a clear limitations section 

Many thanks once again for taking the time to review this article. 

Round 2

Reviewer 1 Report

The authors have improved the manuscript by better describing the nurse service and explaining their approach to understanding how persons using the service felt about healthcare and the benefits of a homeless shelter-based clinic.  While this is not a new concept in other countries, it seems it may have some value in the context of UK health care delivery for those interested in health policy to consider outreach to the homeless population.

Author Response

Dear reviewer

Many thanks for your time and kind attention to our article

Reviewer 2 Report

Thank you for addressing my concerns. Your article is worthy of publication. Good luck!

Author Response

(The authors gave the same response as above.)
